# Increased Risk for Atrial Alternans in Rabbit Heart Failure: The Role of Ca^2+^/Calmodulin-Dependent Kinase II and Inositol-1,4,5-trisphosphate Signaling

**DOI:** 10.3390/biom14010053

**Published:** 2023-12-30

**Authors:** Giedrius Kanaporis, Lothar A. Blatter

**Affiliations:** Department of Physiology & Biophysics, Rush University Medical Center, Chicago, IL 60612, USA; giedrius_kanaporis@rush.edu

**Keywords:** heart failure, atria, alternans, calcium, Ca^2+^/calmodulin-dependent kinase II, arrhythmia, inositol-trisphosphate

## Abstract

Heart failure (HF) increases the probability of cardiac arrhythmias, including atrial fibrillation (AF), but the mechanisms linking HF to AF are poorly understood. We investigated disturbances in Ca^2+^ signaling and electrophysiology in rabbit atrial myocytes from normal and failing hearts and identified mechanisms that contribute to the higher risk of atrial arrhythmias in HF. Ca^2+^ transient (CaT) alternans—beat-to-beat alternations in CaT amplitude—served as indicator of increased arrhythmogenicity. We demonstrate that HF atrial myocytes were more prone to alternans despite no change in action potentials duration and only moderate decrease of L-type Ca^2+^ current. Ca^2+^/calmodulin-dependent kinase II (CaMKII) inhibition suppressed CaT alternans. Activation of IP_3_ signaling by endothelin-1 (ET-1) and angiotensin II (Ang II) resulted in acute, but transient reduction of CaT amplitude and sarcoplasmic reticulum (SR) Ca^2+^ load, and lowered the alternans risk. However, prolonged exposure to ET-1 and Ang II enhanced SR Ca^2+^ release and increased the degree of alternans. Inhibition of IP_3_ receptors prevented the transient ET-1 and Ang II effects and by itself increased the degree of CaT alternans. Our data suggest that activation of CaMKII and IP_3_ signaling contribute to atrial arrhythmogenesis in HF.

## 1. Introduction

In heart failure (HF) structural and functional abnormalities of the heart result in inadequate cardiac output. HF is a leading cause of death in developed countries and its prevalence steadily increases with an aging population. HF greatly increases the probability to develop cardiac arrhythmias including atrial fibrillation (AF). The prevalence of AF in HF patients is very high and is estimated to be around 40% [1]. Unfortunately, the presence of AF in HF raises the risk of adverse outcomes [2]. The mechanisms linking HF to AF are not well understood. Here we investigated Ca^2+^ signaling and electrophysiological disturbances at the single cell level to identify likely candidates for mechanisms that contribute to the higher risk of atrial arrhythmias in HF. The study was performed on rabbit atrial myocytes isolated from normal and failing hearts. In our rabbit HF model HF was induced by combined volume and pressure overload which led not only to ventricular dysfunction but also to atrial structural and functional remodeling [3,4]. We demonstrated that atrial myocytes from HF hearts were larger in size and exhibited larger CaTs, increased SR Ca^2+^ load, higher incidence of spontaneous Ca^2+^ waves, have altered Ca^2+^ removal by Na^+^/Ca^2+^ exchange and a lower density of mitochondria [3,4]. Furthermore, HF atrial myocytes had enhanced inositol-1,4,5-trisphosphate receptor (IP_3_R) dependent Ca^2+^ signaling. Here, as an indicator of increased arrhythmogenicity we looked at the development of atrial alternans. At the cellular level, cardiac alternans is defined as beat-to-beat alternations in contraction amplitude, action potential duration and Ca^2+^ transient (CaT) amplitude at constant stimulation frequency. Alternans is a recognized risk factor for cardiac arrhythmia including AF and is known to precede arrhythmic events [5,6,7].

Ca^2+^ signaling in atrial myocytes has many unique properties compared to ventricular cells. An important difference relates to the expression levels of the IP_3_R Ca^2+^ release channels [8,9,10]. It was hypothesized that upon IP_3_ binding Ca^2+^ release from the sarcoplasmic reticulum (SR) through IP_3_ receptors (IP_3_R) may lower the threshold for Ca^2+^-dependent activation of ryanodine receptor SR Ca^2+^ release channels or directly contribute to the magnitude of Ca^2+^ transients. Also, enhanced IP_3_ signaling was reported in several cardiac pathologies such as hypertrophy, HF [11] and AF [12,13]. In agreement with these findings, our previous study also demonstrated that in rabbits with induced HF, IP_3_ signaling in atria is elevated and release of Ca^2+^ from the SR through IP_3_Rs (IP_3_ induced Ca^2+^ release, IICR) significantly contributes to the magnitude of the CaT [3]. Taken together, these findings suggest that IP_3_-mediated Ca^2+^ signaling plays an important role in cardiac pathologies, however the specific ramifications for atrial function or its role in the progression of HF are poorly understood.

The goal of this study was to determine potential mechanisms contributing to higher atrial arrhythmogenesis in HF. The activity of Ca^2+^/calmodulin-dependent kinase II (CaMKII) in HF increases [10,14] and CaMKII inhibition reduced the risk for ventricular alternans and arrhythmias in HF hearts [15]. In addition, elevated activity of CaMKII was also demonstrated in patients with postoperative AF [16]. Here we demonstrate that block of CaMKII in atria effectively suppresses development of atrial alternans. The role of IP_3_ signaling in the generation of alternans has not been previously investigated, except of our study [17] where we observed a higher incidence of alternans and spontaneous Ca^2+^ release events in feline atrial myocytes treated with endothelin-1 (ET-1) to stimulate IICR. Here we report that activation of IP_3_ signaling in rabbit atrial myocytes has a dual effect on alternans development. Stimulation with the vasoactive agonists ET-1 and angiotensin II (Ang II) resulted in acute reduction in CaT amplitude, lower SR Ca^2+^ load and lowered risk for alternans. However, prolonged exposure to ET-1 and Ang II led to enhanced SR Ca^2+^ release and increased degree of atrial alternans.

## 2. Materials and Methods

### 2.1. Ethical Approval

All aspects of animal husbandry, animal handling, anesthesia, surgery, and euthanasia were fully approved by the Institutional Animal Care and Use Committee (IACUC protocol 22-027; approved 9 May 2022) of Rush University Chicago and comply with the National Institutes of Health’s Guide for the Care and Use of Laboratories Animals.

### 2.2. Heart Failure Model

Nonischemic HF was induced in male New Zealand White rabbits (Envigo, Indianapolis, IN, USA, and Charles River, Wilmington, MA, USA) by combined volume and pressure overload [3,18]. Volume overload was induced by surgical creation of aortic valve insufficiency, followed 2–4 weeks later by induction of pressure overload by abdominal aortic banding. Surgeries were conducted while the rabbits were under isoflurane inhalation anesthesia after premedication with a combination of ketamine, xylazine and acepromazine. HF developed gradually (over 5–8 months, monitored by serial echocardiography) and was characterized by depressed systolic ventricular function. Atrial cells were isolated from hearts that showed marked left-ventricular dilation (54 ± 5% increase in left ventricular diastolic internal dimension) and systolic dysfunction (decrease of left-ventricular fractional shortening by 37 ± 4%). This HF rabbit model has been extensively studied in ventricular and, to some extent atrial myocytes and has been previously characterized at structural, molecular, Ca^2+^ handling, and electrophysiological levels [3,18,19,20]. In this study 19 HF rabbits and 87 normal rabbits were used.

### 2.3. Myocyte Isolation

For isolation of left atrial myocytes rabbits were anaesthetized with an intravenous injection of sodium pentobarbital (100 mg/kg), together with heparin (1000 U/kg) 15 min before thoracotomy. The depth of the anesthesia was evaluated by foot pinch or checking corneal reflexes. Hearts were excised and washed in cold Ca^2+^-free Tyrode solution (in mM): 135 NaCl, 5 KCl, 10 D-Glucose, 5 HEPES; 5 Na-HEPES, 1 MgCl_2_, 1000 U/l Heparin; pH 7.4 (adjusted with 1N HCl). All chemicals and reagents were from Sigma-Aldrich (St. Louis, MO, USA), unless stated otherwise. Hearts were mounted on a Langendorff apparatus and the aorta stub was retrogradely perfused providing coronary perfusion of the heart. After an initial 5–10 min perfusion with oxygenated Ca^2+^-free Tyrode solution at 37 °C the heart was perfused for ~20 min (37 °C) with oxygenated minimal essential medium Eagle MEM solution (Joklik’s modification, Sigma-Aldrich, product #M0518) supplemented with 20 µM Ca^2+^ and 22.5 µg/mL Liberase TH (Roche Diagnostic Corporation, Indianapolis, IN, USA), 2 mM sodium pyruvate, 10 mM taurine, 10 mM HEPES, 10 mM Na-HEPES, 23.8 mM NaHCO_3_, 50,000 U/L penicillin, 50 mg/L streptomycin, and 40 U/L insulin; pH 7.4 (adjusted with 1N HCl). The left atrium was dissected from the heart and minced, filtered and washed in MEM solution with 50 µM Ca^2+^ and 1% bovine serum albumin added. Isolated cells were washed and kept in MEM solution with 50 µM Ca^2+^ at room temperature (20–24 °C) and were used within 8 h after isolation.

### 2.4. Patch Clamp Experiments

The external Tyrode solution was composed of (in mM): 135 NaCl, 5 KCl, 2 CaCl_2_, 1 MgCl_2_, 10 HEPES, 10 D-glucose; pH 7.4 with NaOH. Patch clamp pipettes (1.5–3 MΩ filled with internal solution; see below) were pulled from borosilicate glass capillaries (WPI, Sarasota, FL, USA) with a horizontal puller (model P-97; Sutter Instruments, Novato, CA, USA). Electrophysiological signals were recorded from single atrial myocytes in the whole-cell ruptured patch clamp configuration using an Axopatch 200A patch clamp amplifier, the Axon Digidata 1440A interface and pCLAMP 10.7 software (Molecular Devices, Sunnyvale, CA, USA). AP recordings were low-pass filtered at 5 kHz and digitized at 10 kHz. All patch clamp experiments were performed at room temperature (20–24 °C).

For current-clamp and AP-clamp experiments pipettes were filled with an internal solution containing (in mM): 130 K^+^ glutamate, 10 NaCl, 10 KCl, 0.33 MgCl_2_, 4 MgATP, and 10 HEPES with pH adjusted to 7.2 with KOH. The internal solution was filtered through 0.22-μm pore filters. For AP measurements the whole-cell ‘fast’ current clamp mode of the Axopatch 200A was used and APs were evoked by 4 ms stimulation pulses with a magnitude ~1.5 times higher than AP activation threshold. Membrane potential (V_m_) measurements were corrected for a junction potential error of −10 mV.

For AP clamp experiments voltage commands in form of atrial APs were generated from averages of APs recorded from three individual atrial cells paced at 1.3 Hz and exhibiting CaT alternans [21]. Two voltage commands were generated: AP_CaT_Large_ representing APs recorded during large alternans CaTs, and AP_CaT_Small_, the AP-waveform observed during small alternans CaTs. Reflecting the specific AP shape AP_CaT_Large_ is also referred to as AP_Narrow_ (or AP_N_), and AP_CaT_Small_ as AP_Wide_ (AP_W_).These two distinct AP waveforms were used to generate three pacing protocols: (1) N-N-protocol consisting of series of consecutive AP_CaT_Large_/AP_N_ waveforms; (2) W-W-protocol consisting of series consecutive AP_CaT_Small_/AP_W_ waveforms, and (3) N-W-alternans AP protocol consisting of series of AP_CaT_Large_-AP_CaT_Small_ pairs (...N-W-N-W...sequence) thus mimicking AP alternans. Stimulation frequency was modified by changing diastolic intervals between voltage commands.

For L-type Ca^2+^ current (I_Ca_) measurements patch clamp electrodes were filled with internal solution containing (in mM): 130 Cs^+^ glutamate, 10 NaCl, 10 CsCl, 0.33 MgCl_2_, 4 MgATP, and 10 HEPES with pH adjusted to 7.2 with CsOH. The external solution contained (in mM): 135 Na^+^ glutamate, 4 CsCl, 2 CaCl_2_, 1 MgCl_2_, 10 HEPES, 10 D-glucose; pH 7.4 with NaOH. I_Ca_ was measured in atrial cells during 200 ms voltage steps ranging from −70 mV to +50 mV (with 5 mV and 10 mV increments). Stimulation steps were preceded by 50 ms pre-pulses to −60 mV to inactivate Na^+^ current. Cells were held at −90 mV resting potential and between recordings were pre-stimulated by three steps to 0 mV at 1 Hz to ensure same conditions (SR Ca^2+^ load) during every stimulation step.

### 2.5. [Ca^2+^] Measurements

For cytosolic [Ca^2+^] ([Ca^2+^]_i_) measurements in field stimulation experiments atrial myocytes were loaded with 5 μM Cal520/AM (AAT Bioquest, Sunnyvale, CA, USA) or 5 μM Indo-1/AM (Thermo Fisher Scientific, Waltham MA, USA) in the presence of 0.05% Pluronic F-127 (Thermo Fisher Scientific, Waltham MA, USA) for 20–30 min at room temperature, and then twice washed for 10 min in Tyrode solution to allow for de-esterification of the dye. For field stimulation experiments, glass coverslips were coated with 1 mg/mL laminin to increase adhesion of the cells. CaTs were triggered by electrical field stimulation with a pair of platinum electrodes. The electrical stimulus was set at a voltage ∼50% greater than the threshold to induce myocyte SR Ca^2+^ release. During the course of experiments cells were continuously superfused with Tyrode solution. Fluo-4 (used in AP-clamp experiments, see below) or Cal520 fluorescence was excited at 485 nm with a Xe arc lamp and signals were collected at 515 nm using a photomultiplier tube. Background-subtracted fluorescence emission signals (F) were normalized to resting fluorescence (F_0_) recorded under steady-state conditions at the beginning of an experiment, and changes of [Ca^2+^]_i_ are presented as changes of F/F_0_. Indo-1 fluorescence was excited at 357 nm (Xe arc lamp) and emitted cellular fluorescence was recorded simultaneously at 410 nm (F_410_) and 485 nm (F_485_) with photomultiplier tubes. F_410_ and F_485_ signals were background subtracted and changes of [Ca^2+^]_i_ are expressed as changes in the F_410_/F_485_ ratio. Data recording and digitization were achieved using the Axon Digidata 1440A interface and pCLAMP 10.7 software. Fluorescence signals were low-pass filtered at 30 Hz.

For [Ca^2+^]_i_ measurements during AP-clamp experiments 100 µM Fluo-4 pentapotassium salt (Thermo Fisher Scientific, Waltham MA, USA) was added to the internal solution.

For SR [Ca^2+^] ([Ca^2+^]_SR_) measurements atrial myocytes were loaded with 5 μM of the low-affinity Ca^2+^ dye Cal520N/AM (AAT Bioquest, Sunnyvale, CA, USA) for 60 min at 37 °C in the presence of 0.05% Pluronic F-127. Cal520N fluorescence was excited at 485 nm with a Xe arc lamp and signals were collected at 515 nm using a photomultiplier tube. Background-subtracted fluorescence emission signals (F) were normalized to resting fluorescence (F_0_) recorded in field-stimulated cells at 0.5 Hz. Changes of [Ca^2+^]_SR_ are presented as changes of F/F_0._

### 2.6. CaT Alternans

CaT alternans was induced by incrementally increasing the pacing frequency until stable alternans was observed. The degree of CaT alternans was quantified as the alternans ratio (AR): AR = 1 − [Ca^2+^]_i,Small_/[Ca^2+^]_i,Large_, where [Ca^2+^]_i,Large_ and [Ca^2+^]_i,Small_ are the amplitudes of the large and small amplitude CaTs of a pair of alternating CaTs. By this definition AR values fall between 0 and 1, where AR = 0 indicates no CaT alternans and AR = 1 indicates a situation where SR Ca^2+^ release is completely abolished on every other beat. CaTs were considered alternating when the beat-to-beat difference in CaT amplitude exceeded 10% (AR > 0.1) [22]. The amplitude of a CaT was measured as the difference in F/F_0_ (ΔF/F_0_) measured immediately before the stimulation pulse and at the peak of the CaT.

### 2.7. Drugs

2 mM KN-93, 2 mM KN-92 (Sigma-Aldrich, St. Louis, MO, USA), 1 mM GSK2833503A (Tocris/Bio-Techne, Minneapolis, MN, USA), and 5 mM 2-Aminoethyl diphenylborate (2-APB, Sigma-Aldrich, St. Louis, MO, USA) stock solutions were prepared in DMSO. 0.5 mM stock solution of Angiotensin II and 0.1 mM stock of Endothelin-1 (both from Enzo Life Sciences, Farmingdale, NY, USA) and 1 mM of myristoylated autocamtide-2-related inhibitory peptide (AIP) (Tocris/Bio-Techne, Minneapolis, MN, USA) were prepared in deionized water. Stock solutions were diluted to final concentration in external solutions. Corresponding amounts of DMSO were added to control solutions (final DMSO concentration ≤ 0.1%).

### 2.8. Data Analysis and Presentation

Results are presented as individual observations or as mean ± SEM, n represents the number of individual cells and N is the number of animals. Statistical difference between multiple groups was evaluated using Šídák’s mixed-effects multiple comparisons test for unpaired data or with Tukey’s mixed-effects multiple comparisons test for paired data sets. Two groups of unpaired data were compared with Welch *t*-test. Paired *t*-test was used to compare two groups of paired data. Statistical analysis was performed with GraphPad Prism 9.0 (San Diego, CA, USA). Differences were considered significant at *p* < 0.05.

## 3. Results

### 3.1. Alternans Is Enhanced in Atrial Myocytes from Failing Hearts

We investigated if atrial myocytes isolated from HF hearts are more prone to develop proarhythmic CaT alternans. Atrial myocytes from normal and HF hearts were voltage clamped with AP-shaped voltage commands (AP voltage clamp; Figure 1). Cells were subjected to three AP pacing protocols (details given in Methods section): N-N- and W-W-protocols and the N-W AP alternans protocol. The pacing frequency was stepwise increased from 0.74 to 1.85 Hz. The fraction of cells revealing alternans at a given frequency and the degree of alternans quantified as alternans ratio (AR) were determined. With increasing pacing rates the fraction of cells revealing CaT alternans and the AR increased. HF atrial myocytes clearly exhibited a higher degree of alternans over the entire range of pacing frequencies, however there were quantitative differences between protocols. The most pronounced effect (and the largest difference between control and HF cells) was achieved with the N-W AP alternans protocol (Figure 1A), followed by the W-W protocol (Figure 1B). While the differences of average ARs between normal and HF myocytes was obvious, it was not statistically significant for all pacing rates due to the relative broad spread of ARs in the control group. In contrast, with the N-N protocol (Figure 1C) the stimulation rate-dependence of the AR is comparably flat and the difference between normal and HF atrial myocytes is small, however at higher stimulation rates the same tendency of higher degree of alternans in HF atrial myocytes was observed. The results from control myocytes are consistent with our earlier report [21] that alternans elicited with the N-N protocol tends to have a higher pacing threshold and was less pronounced.

As shown here and previously, inducibility and degree of CaT alternans is strongly modulated by AP morphology [21,23,24]. We analyzed APs from control and HF atrial myocytes recorded at 1 Hz stimulation rate in the absence of alternans (in order to avoid any, in this context undesired feedback effects of CaT alternans on AP morphology) [25]. AP analysis in normal (N/n = 23/28) and HF atrial myocytes (N/n = 5/7) demonstrated that APs are essentially identical and no significant differences in AP duration at 50 and 90 percent repolarization were detected (Figure 2A). This observation and the findings that the degree of alternans is enhanced in AP-clamp experiments using an identical AP shape (Figure 1) suggest that HF promotes alternans through other mechanisms than alterations in AP morphology.

Heart failure was shown to lead to L-type Ca^2+^ current (I_Ca_) remodeling in ventricular and atrial myocytes [26,27,28] and we found that magnitude and kinetics of I_Ca_ were profoundly affected by CaT alternans [21]. Therefore, we compared I_Ca_ properties in normal and HF atrial myocytes. However, we found that peak I_Ca_ recorded at 0 mV was only moderately decreased from 6.60 ± 1.56 pA/pF (N/n = 4/10) in normal atrial myocytes to 5.76 ± 1.71 pA/pF in atrial cells obtained from HF hearts (N/n = 7/16; *p* = 0.2108; Welch t test; Figure 2B). Voltage dependence of peak I_Ca_ also was not affected by HF (Figure 2(Bb)). Consequently, it is unlikely that changes in I_Ca_ are a major contributor to higher alternans risk in HF.

### 3.2. Enhanced CaMKII Activity Increases Risk for Atrial Alternans

It is well established that in HF remodeling leads to increased activity of CaMKII including in the atria atria [10,14,15,29,30,31]. To determine if manipulation of CaMKII activity could affect the propensity for CaT alternans we inhibited CaMKII in normal atrial myocytes. Figure 3(Aa) shows CaTs recorded in the same field-stimulated atrial cell before and after application of CaMKII inhibitor KN-93 (1 µM). In the presence of KN-93 the degree of CaT alternans was significantly smaller indicating that CaMKII activity promoted development of alternans. In contrast, KN-92 (1 µM), an inactive analog of KN-93 that does not affect CaMKII activity, had no effect on the severity of CaT alternans (Figure 3(Aa) bottom traces). These data are summarized in Figure 3(Ab) showing that application of KN-93 reduced AR from 0.55 ± 0.16 to 0.18 ± 0.12 (N/n = 4/10; *p* < 0.0001; paired t test), while KN-92 had no significant effect on the degree of CaT alternans (AR was 0.51 ± 0.17 in control and 0.47 ± 0.26 in KN-92; N/n = 4/8; *p* = 0.7028).

In addition to KN-93, we also used the specific CaMKII inhibitor autocamtide-2-related inhibitory peptide (AIP; Figure 3B). In this case alternans was induced by incrementally increasing pacing frequency separately in control cells and in cells incubated for >30 min with the cell permeable form of AIP (1 µM). Measurements in control and in AIP were carried out on the same day in alternating fashion to ensure the same experimental conditions. Figure 3(Ba) shows original CaTs in control and in AIP treated atrial cells stimulated at 1.6 Hz. While in a control myocyte pronounced CaT alternans was observed, at the same pacing rate CaT alternans was essentially absent in an AIP treated cell. Figure 3(Bb) shows mean CaT ARs recorded at pacing frequencies between 0.5 and 2 Hz in control cells (N/n = 8/35) and cells preincubated with AIP (N/n = 8/24). Figure 3(Bc) shows a subset of data that include only cells exhibiting alternans (AR > 0.1), while Figure 3(Bd) shows the fraction of cells exhibiting alternans in control and in presence of AIP, respectively, as a function of pacing frequency. The data show that in presence of AIP the propensity of CaT alternans is decreased across the entire range of stimulation frequencies tested and the suppression of CaT alternans by AIP is consistent with the results obtained with KN-93. Taken together our data clearly demonstrates that inhibition of CaMKII reduces the risk for alternans in atrial myocytes, suggesting that increased CaMKII activity in HF is a potential mechanism leading to higher propensity of alternans.

### 3.3. Activation of IP_3_ Signaling in Atrial Myocytes Affects Alternans Development and CaT Properties

Previously we have reported that atrial IP_3_ signaling is significantly enhanced in our HF model [3]. Therefore, we tested if stimulating IP_3_ signaling in normal atrial myocytes affects pacing induced CaT alternans. To increase IP_3_ production in atrial myocytes we stimulated cells with 100 nM ET-1 or 500 nM Ang II. We have demonstrated previously that ET-1 and Ang II increase IP_3_ levels in cardiac myocytes [3]. Application of ET-1 or Ang II had a dual effect on the degree of CaT alternans. First, an acute reduction of CaT AR was observed with a maximal effect occurring within the first 0.5–2 min of drug application (Figure 4A for ET-1 and Figure 4B for Ang II). Subsequently, CaT alternans reappeared and cells eventually displayed alternans with higher AR than in control. Exposure to ET-1 or Ang II was followed by 5 min washout which had no statistically significant effect on CaT AR, suggesting that ET-1 and Ang II application leads to a modulation of intracellular Ca^2+^ signaling that is not reversed during a 5 min washout period. During the acute phase of the ET-1 effect mean CaT AR declined from 0.38 ± 0.16 to 0.03 ± 0.01 (Figure 4(Ab)). However, subsequently alternans rebounded to a higher degree and after 10 min mean AR was 0.61 ± 0.21 (N/n = 4/8; Figure 4(Ab)). Similarly, application of Ang II reduced CaT AR from 0.50 ± 0.13 to 0.20 ± 0.23 acutely, and after 10 min of exposure to Ang II CaT AR rose to 0.66 ± 0.18 (N/n = 4/9; Figure 4(Bb)). Our results show that prolonged exposure to ET-1 and Ang II increased the degree of CaT alternans which mimicks our observations in HF where cells demonstrated a higher risk and a higher degree of CaT alternans (Figure 1) and had enhanced IP_3_ signaling and elevated [IP_3_] levels [3].

Next, we investigated underlying mechanisms leading to CaT AR modulation during ET-1 and Ang II application. First, we tested the effects of these IICR activators on CaT properties in normal atrial myocytes paced at 0.5 Hz frequency (Figure 5). Application of ET-1 (100 nM; Figure 5A) or Ang II (500 nM; Figure 5B) led initially to a transient decrease of the CaT amplitude (within the first 2 min of drug application), followed by a recovery of the CaT amplitude to levels similar or even higher than in control conditions. Mean CaT ΔF/F_0_ in control was 3.29 ± 1.11 and was reduced to 2.10 ± 0.77 after application of ET-1 (N/n = 4/13, *p* < 0.0001; Tukey’s test; Figure 5(Ab)) and later increased to 3.76 ± 1.55 (*p* = 0.0008). Similar results were observed during the application of Ang II (Figure 5(Bb)), where CaT amplitude decreased transiently from 2.24 ± 0.53 in control to 1.59 ± 0.43 (N/n = 5/9, *p* = 0.0005) and then increased again to 2.91 ± 1.01 (*p* = 0.0033). Diastolic [Ca^2+^] levels, however, were not affected by ET-1 and Ang II.

To further investigate the observed transient decrease in CaT amplitude during the acute exposure to ET-1 and Ang II, we investigated the effect of ET-1 on SR Ca^2+^ load. [Ca^2+^]_SR_ was monitored with the low affinity Ca^2+^ dye Cal520N entrapped in the SR in atrial myocytes paced at 0.5 Hz. Figure 6A shows that application of ET-1 leads to acute reduction in [Ca^2+^]_SR_ which later recovered over a time course of several minutes. The average amplitude of beat-to-beat depletions of [Ca^2+^]_SR_ in the presence of ET-1 decreased from ΔF/F_0_ 0.198 ± 0.099 in control to 0.071 ± 0.025 (N/n = 2/7; *p* = 0.0170; Tukey’s test; Figure 6B) and recovered to 0.192 ± 0.049 (N/n = 2/6) after 10 min of ET-1 exposure. The changes of [Ca]_SR_ depletion amplitudes were paralleled by a transient decrease of diastolic [Ca^2+^]_SR_ from 1.00 ± 0.01 to 0.89 ± 0.07 (*p* = 0.0095, Figure 6C) which recovered back to 0.94 ± 0.06 after 10 min. Our results indicate that activation of IP_3_ signaling has a dual effect on atrial Ca^2+^ handling. First, enhanced Ca^2+^ release from SR due to IP_3_ receptor activation leads to SR Ca^2+^ depletion, lower [Ca^2+^]_SR_ and subsequently lower CaT amplitudes, but also concomitantly to a lower propensity of CaT alternans development. Over time, however, SR Ca^2+^ load is restored and CaT amplitude recovered, but so does the risk for CaT alternans.

### 3.4. Effect of IP_3_R and TRP Channel Inhibition on CaT Alternans

We also tested if inhibition of IP_3_ receptors alone would affect the development of alternans in normal atrial myocytes. Application of the IP_3_ receptor blocker 2-APB (5 µM) consistently increased the degree of CaT alternans (Figure 7A,B) and this effect was reversible after washout of 2-APB. In the presence of 2-APB the mean CaT AR increased from 0.26 ± 0.17 to 0.69 ± 0.19 (N/n = 5/10, *p* < 0.0001; Tukey’s multiple group comparison test) and during washout returned to 0.19 ± 0.20 (N/n = 3/8 *p* < 0.0001, Figure 7B). 2-APB had no effect on CaT amplitude (recorded at 0.5 Hz pacing in the absence of CaT alternans, Figure 7(Ca)) but resulted in higher diastolic [Ca^2+^]_i_ (Figure 7(Cb)). Application of 2-APB in cells pretreated with Ang II led to an additional increase in CaT AR (Figure 7D) indirectly suggesting that Ang II and 2-APB might increase alternans by different mechanisms as their effects were additive.

2-APB has been shown to block transient receptor potential (TRP) channels [32]. Furthermore, Ang II and ET-1 stimulation has also been suggested to activate TRP channels [33]. Therefore, we tested whether TRP channel inhibition affected CaT alternans or the effect of 2-APB on CaT AR. For this purpose we used the TRPC channel blocker GSK2833503A (200 nM) which blocks TRPC3 and TRPC6 channels [34], the two most abundant TRPC channels in atria. GSK2833503A had no effect on the degree of alternans (Figure 7E; N/n = 4/7; *p* = 0.8534) and did not abolish the increase in AR in response to 2-APB.

Next, we applied Ang II to 2-APB pretreated atrial myocytes to test if the Ang II induced effects on CaT alternans are due to IICR. Figure 8A shows that while block of IP_3_ receptors alone increased CaT AR, it also completely abolished the acute effect of Ang II on CaT AR, i.e., in the presence of 2-APB application of Ang II led immediately to an increase in AR and failed to generate a transient AR decrease. 2-APB also prevented the acute transient CaT amplitude decrease during exposure to Ang II in cells paced at 0.5 Hz (N/n = 3/12; Figure 8B).

Additionally, we tested whether TRP channel inhibition could be involved in the CaT amplitude modulation by ET-1. However, GSK2833503A did not alter the effect of ET-1 on CaT amplitude (N/n = 3/8, Figure 8C), suggesting that the effects of ET-1 and Ang II are independent of TRPC activity.

Our data suggest that IICR could be protective as IP_3_R inhibition by 2-APB enhanced the degree of CaT alternans and abolished a transient CaT AR decrease during Ang II application. However, 2-APB did not prevent an increase of the degree of alternans during prolonged exposure to Ang II, suggesting that Ang II increases the risk for alternans also through pathways independent from IICR.

## 4. Discussion

### 4.1. HF Remodeling Leads to Higher Risk of Proarrhythmic Alternans

HF is a leading cause of death in developed countries and its prevalence steadily rises with an aging population. In addition to ventricular arrhythmias that can lead to sudden death, HF also increases the propensity of atrial arrhythmias, including AF [1]. The presence of AF in HF increases the risk of complications and mortality [2,35]. The mechanisms linking HF and AF are multifactorial and complex, and are not fully understood. One of the factors contributing to AF is structural remodeling of the atria, leading to dilation accompanied by an increase in atrial fibrosis [36,37]. Tissue fibrosis results in slower and heterogenous AP propagation enhancing the risk of re-entry. In addition, HF was shown to increase excitation of pulmonary vein cardiomyocytes [38] which act as a frequent trigger for atrial arrhythmias. HF-dependent changes in Ca^2+^ cycling were mostly investigated in ventricle, but there is also strong evidence that atrial Ca^2+^ signaling in HF is altered [39]. Typical changes reported are increased SR Ca^2+^ load and enhanced SR Ca^2+^ leak, increased Na^+^/Ca^2+^ exchange [40] while I_Ca_ is reduced. Increase in SR Ca^2+^ content and SR Ca^2+^ leak is associated with higher incidence of spontaneous Ca^2+^ releases from SR [3,4] that can lead to delayed afterdepolarizations.

This study focuses on cellular changes leading to a higher risk for atrial CaT alternans, which is an indicator for enhanced arrhythmogenesis at the single myocyte level. Clinical studies have demonstrated that atrial alternans, similar to ventricular T-wave alternans, precede AF episodes and can be used to predict vulnerability to AF [41,42]. In atrial myocytes isolated from failing hearts compared to normal myocytes we observed a higher degree of alternans as well as lower pacing threshold for alternans induction (Figure 1). These differences between HF and normal atrial myocytes were observed in voltage-clamped cells. These results indicate that in HF remodeling results in changes of cell signaling at the single myocyte level that are independent of action potential morphology and predispose to a higher risk of alternans.

Observations how HF affects atrial electrophysiology have been inconsistent as prolongation [43], shortening [27,44] or no change [45] of action potential duration have been reported. We did not find any significant change in AP morphology in our model (Figure 2A). The decrease in atrial I_Ca_ in HF is commonly observed [26,27,28,45] and is associated with a reduced expression of the channel or atrial transverse tubule (t-tubule) remodeling [26]. In accord with these reports, our results show a slight reduction in atrial I_Ca_ in HF (Figure 2B). However, the decrease in I_Ca_ was moderate and did not change the voltage-dependence of I_Ca_, and thus is unlikely to explain alone the higher risk for alternans in HF atria.

### 4.2. CaMKII Promotes Atrial Alternans

Studies on AF pathophysiology indicate that one of the mechanisms of AF is related to increased Ca^2+^ leak from the SR which can be quantified by the frequency of Ca^2+^ sparks and waves. Enhanced RyR phosphorylation by CaMKII was shown to be one of the main factors contributing to the increased diastolic SR Ca^2+^ leak [46]. Increased CaMKII activity in HF is well documented and plays a key role in cardiac remodeling [14,47]. In HF expression of CaMKIIδ is increased [29] and activity of CaMKII is further enhanced by posttranslational modifications [31]. Most studies on HF and CaMKII focused on ventricle, and experimental and computational studies suggest that higher activity of CaMKII promotes ventricular alternans in HF and other heart pathologies [48,49,50,51]. Here we tested if suppression of CaMKII activity affects development of alternans in atrial myocytes. Inhibition of CaMKII by KN-93 was an efficient way to suppress alternans, while application of KN-92 as a negative control had no effect on alternans (Figure 3). As an alternative method, we preincubated atrial myocytes for at least 30 min with a membrane permeable form of the CaMKII inhibitor AIP. Experiments with AIP showed the same trend as experiments with KN-93, i.e., inhibition of CaMKII reduced the degree of alternans and the fraction of cells that showed alternans at a given pacing frequency. These results provide support for the hypothesis that increased CaMKII activity in HF is a potential contributor to increased alternans propensity.

### 4.3. Modulation of IP_3_ Signaling Pathways Affects Development of Atrial Alternans

While excitation-contraction coupling (ECC) in atrial and ventricular myocytes share similar mechanisms there are also important differences. T-tubules, deep invaginations of the sarcolemmal membrane, allow AP penetration to the interior of the cell and ensure fast and uniform SR Ca^2+^ release in ventricular myocytes. Atrial cells lack or have only a poorly or irregularly developed t-tubule system. As a consequence, L-type Ca^2+^ channels are restricted to the periphery of the cell and thus, AP induced Ca^2+^ release is spatially inhomogeneous and first occurs in subsarcolemmal regions and subsequently propagates to the center of the cell via Ca^2+^ induced Ca^2+^ release (summarized in [52]). Atrial and ventricular myocytes also differ in the expression and activity of IP_3_ receptors [8,9]. While RyRs are the main channels for Ca^2+^ release from the SR and significantly outnumber IP_3_Rs, numerous studies have demonstrated that in atria IICR exerts a positive inotropic effect [8,17]. Furthermore, levels of IP_3_ signaling and expression of IP_3_R were shown to increase in various cardiac pathologies such as cardiac hypertrophy [11], HF [3,11] or AF [13]. While IP_3_ dependent regulation of gene transcription contributing to hypertrophic remodeling of atria and ventricles was clearly demonstrated [11,13,53], the role of enhanced IP_3_ signaling in pathophysiological atrial ECC and arrhythmogenesis is less clear. Our previous study has demonstrated that in our HF model IP_3_ signaling in atria is increased leading to enhanced atrial contractility [3]. Stronger atrial contraction improves ventricular filling resulting in improved ventricular output that (partially) compensates for the deteriorating cardiac performance in HF. However, enhanced IP_3_ signaling also could lead to an adverse increase of the risk of arrhythmias through more frequent spontaneous Ca^2+^ releases or increased SR Ca^2+^ leak consequently triggering delayed after-depolarizations [3]. Here we investigated if activation of IP_3_ signaling by application of ET-1 and Ang II affects the development of CaT alternans in atrial myocytes. ET-1 and Ang II had a dual effect on alternans (Figure 4): at first alternans was suppressed transiently, but during the prolonged exposure to ET-1 or Ang II alternans recovered and after 10 min of treatment exhibited higher ARs than in control. Next, we investigated underlying mechanisms leading to the acute suppression and subsequent increase of alternans. A similar dual effect of application of ET-1 and Ang II was also observed for CaT amplitude. Initially the CaT amplitude was reduced but later recovered to similar or higher levels than in control (Figure 5). Reduction in CaT amplitude was associated with a decrease of SR Ca^2+^ load (Figure 6). Acute suppression of alternans, and decrease of CaT amplitude and [Ca^2+^]_SR_ had a similar time course and the maximal acute effect was reached between 0.5 and 2 min. Our data indicate that acute suppression of alternans by the application of Ang II or ET-1 is due to IICR because block of IP_3_ receptors by 2-APB completely abolished this effect (Figure 8A) as well as prevented the acute decrease in CaT amplitude (Figure 8B). In addition, 2-APB alone also promoted alternans (Figure 7A,B). Therefore, we suggest that activation of Ca^2+^ release through IP_3_R is protective against alternans at least during acute phase of ET-1 or Ang II exposure, while inhibition of IP_3_Rs increases risk and degree of CaT alternans.

2-APB is also known to block TRP channels. However, GSK2833503A, which selectively blocks the most ubiquitous TRP channels in atria TRPC3 and TRPC6 [34], had no effect on CaT parameters (Figure 8C) or alternans (Figure 7E), and did not alter the response to ET-1 (Figure 8C). Therefore, we conclude that effects of 2-APB on alternans are not associated with TRP channel activity.

## 5. Conclusions

The risk for proarrhythmic alternans increases in HF (Figure 1). We demonstrate that intracellular remodeling other than changes in cell electrophysiology (Figure 1 and Figure 2) predispose atrial myocytes to alternans. Enhanced CaMKII activity is one of the potential mechanisms leading to higher risk of CaT alternans (Figure 3). We also explored the potential role of IP_3_ signaling in the development of alternans. Our data demonstrate that activation of IICR could be protective (Figure 4 and Figure 8A). We propose that release of Ca^2+^ through IP_3_Rs leads to lower SR Ca^2+^ load (Figure 6) and possibly lesser RyR sensitivity to cytosolic Ca^2+^ thus decreasing the risk for alternans. Inhibition of IP_3_Rs has an opposite effect, i.e., it increased the degree of alternans and completely prevented the acute decrease of alternans normally induced by Ang II (Figure 7). This effect most likely is due to increased [Ca^2+^]_SR_ and leads to higher cytosolic Ca^2+^ leak indicated by higher diastolic [Ca^2+^]_i_ during IP_3_ receptor inhibition (Figure 7(Cb)).

We identify increased CaMKII activity and enhanced IP_3_ signaling in HF as potential pathways leading to increased atrial arrhythmogenesis. Therefore, we suggest that vasoconstriction agents activating IP_3_ signaling might be linked to a higher risk of atrial arrhythmias. On the other hand, antihypertensive treatments lowering Ang II levels could be beneficial in reducing arrhythmias risk in HF patients. Our data in atrial myocytes also complements findings that CaMKII inhibition reduces the risk of ventricular arrhythmias [14,15,47,50] suggesting that targeting CaMKII signaling has a considerable potential for cardiac therapies.

## Figures and Tables

**Figure 1 biomolecules-14-00053-f001:**
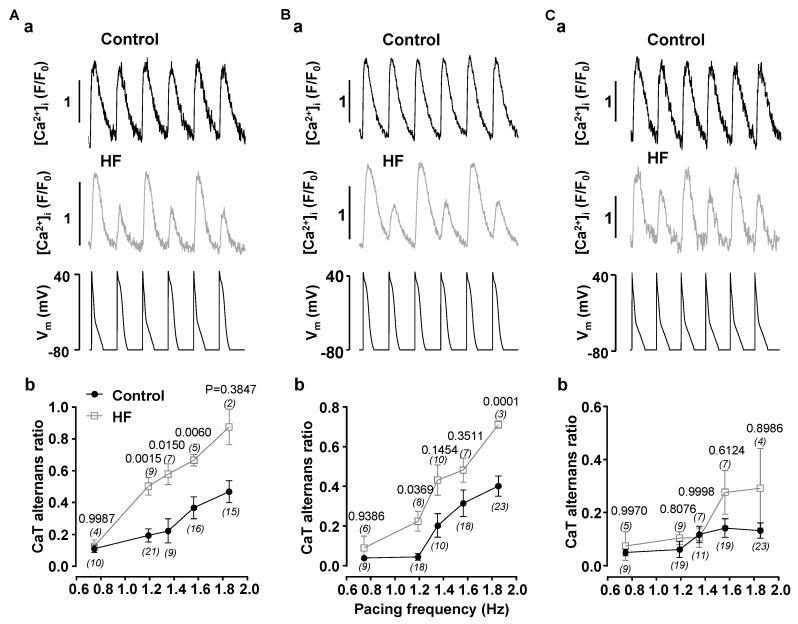
CaT alternans is enhanced in heart failure. CaT traces recorded in AP-clamped atrial myocytes isolated from normal (Control) and HF hearts paced with N-W ((**Aa**), paced at 1.19 Hz), W-W ((**Ba**) paced at 1.19 Hz) and N-N ((**Ca**) paced at 1.56 Hz) pacing protocols. Bottom traces in (**a**) panels show pacing protocols. Mean ± SEM of ARs observed in control and HF atrial myocytes paced at different frequencies with AP-clamp N-W (**Ab**), W-W (**Bb**) N-N (**Cb**) pacing protocols. Data was collected from cells isolated from 19 control and 7 HF rabbits, number of cells for every data point is shown in parentheses. Statistics calculated using Šidàk’s mixed-effects multigroup comparison test.

**Figure 2 biomolecules-14-00053-f002:**
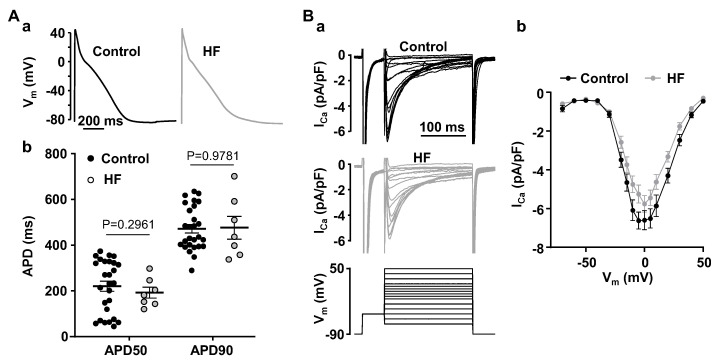
AP morphology and I_Ca_ in normal and HF atrial myocytes. (**A**): (**a**) Atrial APs recorded in normal and HF current clamped myocytes at 1 Hz. (**b**) Mean ± SEM and individual AP durations at 50% and 90% of repolarization in control (N/n = 23/28) and HF (N/n = 5/7) myocytes recorded at 1 Hz pacing. Statistical analysis performed using Welch *t*-test. (**B**): (**a**) Representative I_Ca_ traces in normal and HF atrial myocytes recorded during step stimulation protocol (bottom). (**b**) I-V curves of I_Ca_ recorded in normal (N/n = 4/10) and HF (N/n = 7/16) atrial myocytes. No statistical differences between I_Ca_ amplitude in control and HF myocytes (Šidàk’s test).

**Figure 3 biomolecules-14-00053-f003:**
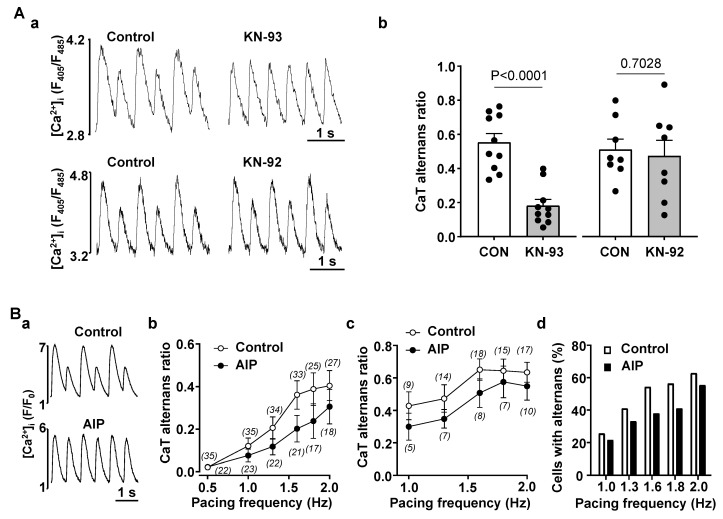
Inhibition of CaMKII activity decreases risk of atrial alternans. (**A**): (**a**) CaT traces recorded in control and after application of CaMKII blocker KN-93 (1 µM; top) and its inactive analogue KN-92 (1 µM; bottom). (**b**) Mean ± SEM and individual CaT ARs recorded in control and after the application of KN-93 and KN-92 (paired *t*-test). (**B**): (**a**) CaT traces recorded in control cells and in cells preincubated with AIP (1 µM) at 1.6 Hz pacing. (**b**) Mean ± SEM CaT ARs observed at various pacing frequencies in control cells and in cells preincubated for ≥30 min with AIP (1 µM). (**c**) Mean ± SEM CaT ARs at different pacing rates in control and AIP-treated cells exhibiting alternans (in this data set non-alternating cells were removed from the analysis. (**d**) Fraction of cells exhibiting alternans at given frequencies in control cells (N/n = 8/35) and cells preincubated with AIP (N/n = 8/24).

**Figure 4 biomolecules-14-00053-f004:**
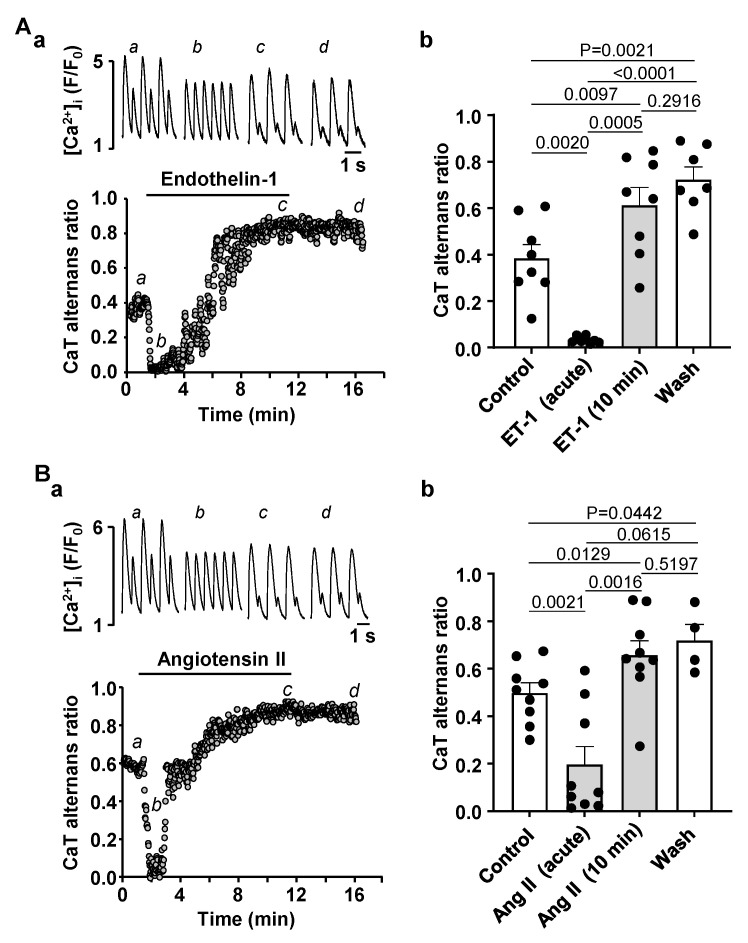
Stimulation of IP_3_ signaling by ET-1 and Ang II has a dual effect on CaT alternans. (**A**): (**a**) CaT ARs recorded over time in control and in the presence of 100 nM ET-1. Top: CaT traces observed at the times marked by the corresponding letter in the bottom graph. (**b**) Mean ± SEM and individual CaT ARs observed in control, after 0.5–2 min of application of ET-1 at the point of maximal effect of ET-1 (acute effect), after 10 min of ET-1 application and following ≥5 min washout of ET-1 (N/n = 3/8; Tukey’s test). (**B**): (**a**) CaT ARs recorded over time in control and in the presence of 500 nM Ang II. Top: CaT traces observed at the times marked by the corresponding letter in the bottom graph. (**b**) Mean ± SEM and individual CaT ARs observed in control, after acute application of 500 nM of Ang II, after 10 min of Ang II application and after ≥5 min washout (N/n = 4/9 Tukey’s test).

**Figure 5 biomolecules-14-00053-f005:**
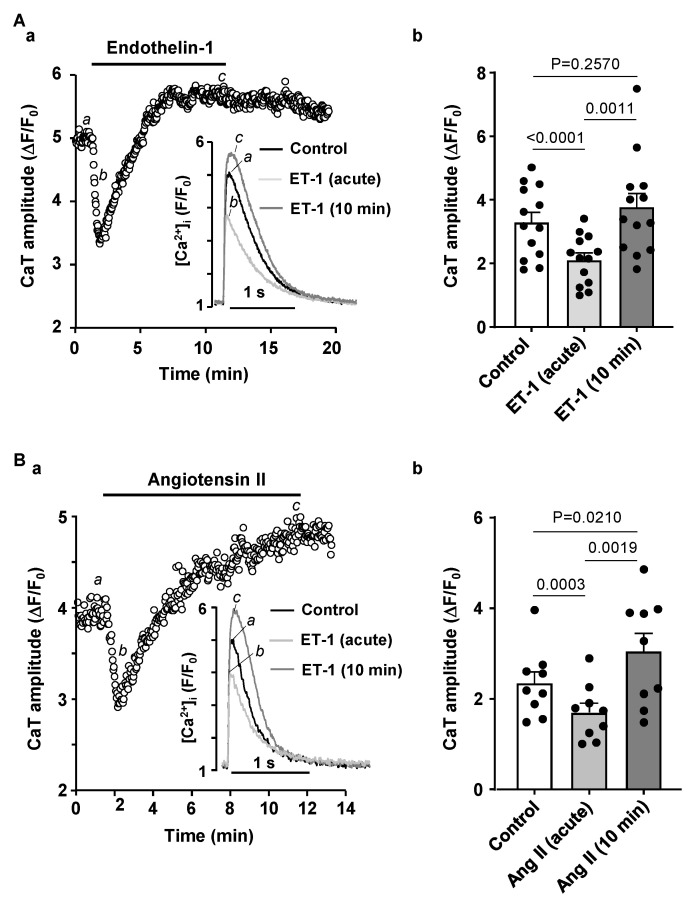
Effect of endothelin-1 and angiotensin II on CaT amplitude. (**A**): (**a**) CaT amplitude recorded over time in control and in the presence of 100 nM ET-1 in atrial cell paced at 0.5 Hz. Inset: Overlay of CaTs recorded in control, after acute and 10 min application of ET-1. (**b**) Mean ± SEM and individual CaT amplitudes in control, maximal transient effect of ET-1 (ET-1 acute) and after 10 min application of 100 nM ET-1 (N/n = 4/13, Tukey’s test). (**B**): (**a**) CaT amplitude recorded over time in control and in the presence of 500 nM Ang II. Inset: Overlay of CaTs recorded in control, after acute and 10 min application of Ang II. (**b**) Mean ± SEM and individual CaT amplitudes in control, after acute and 10 min application of 100 nM Ang II (N/n = 6/9, Tukey’s test).

**Figure 6 biomolecules-14-00053-f006:**
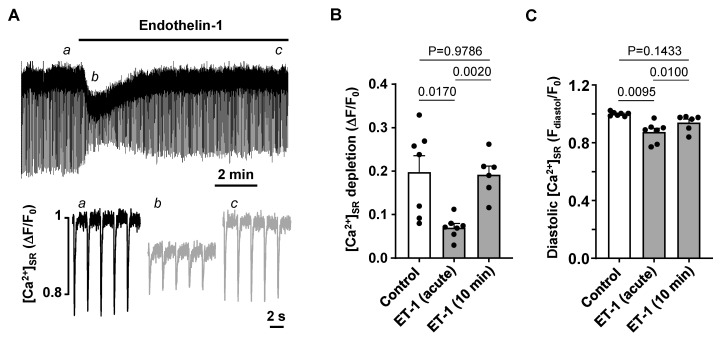
Effect of endothelin-1 on SR Ca^2+^ load. (**A**) [Ca^2+^]_SR_ measured with the low affinity Ca^2+^ indicator Cal520N in control and in 100 nM ET-1 in cells paced at 0.5 Hz. Bottom: depletions of [Ca^2+^]_SR_ recorded at times indicated by the corresponding letter in the top graph. (**B**,**C**): Mean ± SEM and individual depletions of [Ca^2+^]_SR_ (**B**) and diastolic [Ca^2+^]_SR_ (**C**) measured in control, during transient ET-1 effect and after 10 min application of ET-1 (N/n = 2/7). Statistics: Tukey’s test.

**Figure 7 biomolecules-14-00053-f007:**
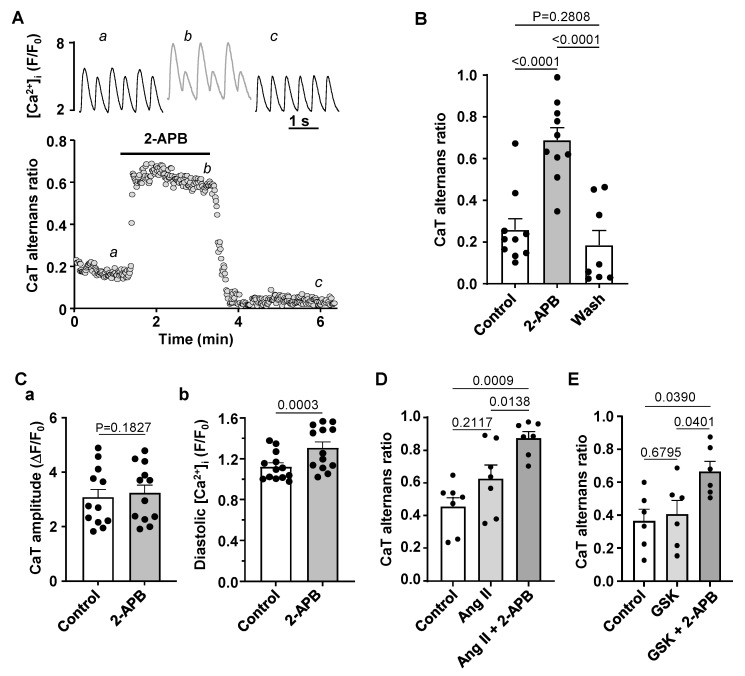
IP_3_ receptor blocker 2-APB enhances CaT alternans. (**A**): Traces of CaTs (top) and CaT ARs recorded over time in control and in presence of 2-APB (5 µM). (**B**): Mean ± SEM and individual CaT ARs observed in control, in 2-APB and after washout of 2-APB (N/n = 4/10, Tukey’s test). (**C**): Mean ± SEM and individual CaT amplitudes (**a**) and diastolic [Ca^2+^]_i_ (**b**) observed in control and in the presence of 2-APB recorded at 0.5 Hz pacing in the absence of alternans (N/n = 3/12, paired *t*-test). (**D**): Application of 2-APB in the presence of Ang II further increases CaT AR (N/n = 2/7; Tukey’s test). (**E**): Mean ± SEM and individual CaT ARs recorded in control and after application of TRPC6/3 channel blocker GSK2833503A (GSK; 200 nM) and simultaneous application of GSK and 2-APB (N/n = 4/7; Tukey’s test).

**Figure 8 biomolecules-14-00053-f008:**
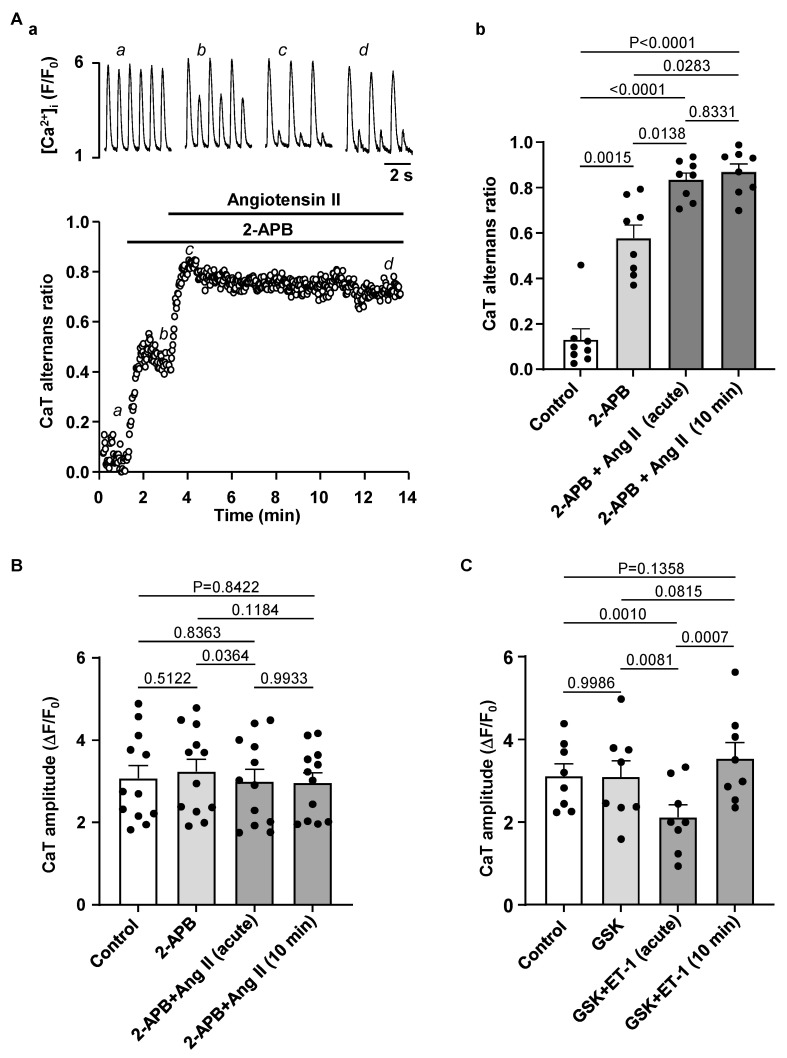
IP_3_R block prevents transient effect of Ang II on CaT alternans and CaT amplitude. (**A**): (**a**) CaTs (top) and CaT ARs (bottom) recorded over time in control, in 5 µM 2-APB and during simultaneous application of 2-APB and 500 nM Ang II. (**b**) Mean ± SEM and individual CaT ARs recorded in control, in 2-APB and in 2-APB together with Ang II after 1 min (time point when typically the transient effect of Ang II is observed under control conditions) and after 10 min of Ang II application (N/n = 3/8, Tukey’s test). (**B**): Application of IP_3_ receptor blocker 2-APB (5 µM) prevented the acute effect of Ang II on CaT amplitude (0.5 Hz pacing, N/n = 3/12, Tukey’s test). (**C**): TRPC6/3 channel blocker GSK2833503A (200 nM) had no effect on CaT amplitude and did not prevent transient decrease of CaT amplitude during ET-1 application (pacing at 0.5 Hz; N/n = 3/8, Tukey’s test).

## Data Availability

The data presented in this study are available on request from the corresponding author.

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
