# Peer review of "Increased Risk for Atrial Alternans in Rabbit Heart Failure: The Role of Ca2+/Calmodulin-Dependent Kinase II and Inositol-1,4,5-trisphosphate Signaling"

_biomolecules, 2023, doi:10.3390/biom14010053_

Round 1

Reviewer 1 Report

Comments and Suggestions for Authors

The paper presents the results of the investigation  the role of Ca modelling in rabbit heart failure models. The research is methodologically robust, and the results are interesting. I have no ethical remarks, as all the investigations on animals seem to be prepared following good practice. I strongly recommend the paper for publication, but I have 2 minor remarks:

1.      As the study is performed on animal model, it should be clearly stated in the title.

2.      I think that the paragraph regarding the possible clinical implications would be valuable.

Author Response

Concern 1: As the study is performed on animal model, it should be clearly stated in the title.

Response: We have changed the title to:

"Increased risk for atrial alternans in rabbit heart failure: the role of Ca2+/calmodulin-dependent kinase II and inositol-1,4,5-trisphosphate signaling".

Concern 2: I think that the paragraph regarding the possible clinical implications would be valuable.

Response: We have added a paragraph to the discussion section.

Reviewer 2 Report

Comments and Suggestions for Authors

The manuscript under review explores the role of calcium signaling, CaMKII and IP3, in the development of alternans in the heart following heart failure. The authors induced heart failure in rabbits and performed further studies in atrial myocytes using patch-clamp study and fluorescence microscopy. They found that myocytes from heart failure hearts are more prone to alternans, alterations in amplitude of cytosolic calcium increases that accompany each action potential (“heart beating”). Furthermore, they established that these alterations in calcium signaling are not due to changes in electrical activity, action potentials, as their amplitude and duration was the same for both normal and heart failure myocytes. Their further study showed that inhibition of CaMKII can suppress alternans, while modulating IP3 receptors with ET-1 and Angiotenis II had dual effects. The authors also demonstrated that another possible player, TRP channels, are not involved. Despite the lack of clear mechanism how CaMKII and IP3 induce calcium alternans in atrial myocytes following heart failure, the study is well implemented and of interest. I have the following suggestions to improve the ms quality:

1. Make a more straightforward labeling of panels in the figures. Like A,B,C,D etc. Current labeling, Aa, Ba, Ca etc, is not easy to following.

2. Fig. 1 Labeling of Aa, Ba, Ca axes should be checked and fixed as it is not consistent. (F/F0 or F405/485).

Comments on the Quality of English Language

English is ok

Reviewer 3 Report

Comments and Suggestions for Authors

   The study investigates calcium signalling and electrophysiological disturbances to identify likely candidates for mechanisms that contribute to the higher risk of atrial arrhythmias in heart failure. The development of atrial alternans was used as an indicator of increased arrhythmogenicity.

   A slowly developing rabbit model of heart failure following aortic valve insufficiency and abdominal aortic banding was utilised following development of LV dilatation and systolic dysfunction. Left atrial cardiomyocytes were isolated and the following experiments carried out at room temperature in cardiomyocytes from control and heart failure animals.

  Overall the study has been carefully carried out, is well described, and was a pleasure to read. I do have some Questions for the authors:

Questions:

1.     The animal model used was of LV failure. What evidence was there that the atrial myocytes were also "failing"?

2.     Were there differences in heart rate between groups? Since alternans was increased at higher stimulation frequencies, might this explain the higher incidence in HF animals?

3.     Has IP3R expression levels been compared between atrial myocytes from Control & HF animals? Was there a difference in IP3R expression between groups?

4.     Figure 1 a. Why was a different pacing frequency shown for the Control & HF representative traces in C? Why is the Ca transient amplitude much smaller in the example shown in panel B Control? Perhaps that cell was not a good choice?

5.     Figure 2. Can you make the connection clearer to the reader between the APs shown in this figure, and those chosen for AP clamped myocytes in Figure 1?

6.     Endothelin-1 and angiotensin II both have a number of subcellular targets that affect calcium handling in cardiomyocytes. How was it determined that the observed effects of the drugs was via the IP3Rs alone?

7.     How was it determined that the Cal520N/AM signal was from the SR, and did not include beat-to-beat changes in mitochondrial calcium?

8.     Figure 3 shows data from Control myocytes only, establishing a link between alternans and CaMKII. However, data is not provided for HF myocytes, although it is suggested that increased CaMKII activity underlies the increased prevalence of alternans in HF myocytes. This should be tested and data provided.

9.     The different scales used for representative Ca transients (e.g. as in Fig. 3 A a) suggest some cells had much higher resting calcium levels than others, perhaps influencing the data. Overall, numbers of hearts/cells per group at each frequency are lower in HF. Why were the calcium transient amplitudes so much larger in Fig 3 B b compared with Fig 3 A a? Doesn't such variability in Ca transient amplitude make comparisons between groups difficult?

10.  Fig 3 B d showed 60% of Control cells had alternans at 2 Hz stimulation frequency. That seems incredibly high to me! Resting heart rate in a rabbit is 140 -180 beats per min. Obviously, room temp affects the calcium handling, but that still seems a very high portion of control cells showing alternans under these experimental conditions.

11.  Section 3.3 showed an acute phase following application of both ET-1 and Ang II. Was this due to a transient difference in concentration once drugs were introduced to the cells? Were cells placed into solutions containing the drug, or were the drugs applied via super-fusion? If the latter, then could different concentration affects explain the acute responses as the drugs wash in?

12.  Figure 5 shows Ca transient amplitude is decreased during acute ET-1 application, without providing information of resting calcium levels. This information would be important to show since Fig 6 suggests SR Ca load is decreased by ET-1. Was there any effect on the re-uptake of Ca into the SR? Was the decay of the Ca transients investigated? Fig 5 also shows a much higher Ca transient amplitude during phase c (late ET-1 application), yet Fig 6 shows the SR release in this late phase of the ET-1 application is not as large as the pre-ET-1 phase. How can this be explained?

13.  Given the hypothesis that alternans is dependent on increased CamKII activity, how do you explain the increased likelihood of alternans occurring at low temperature and increased stimulation frequency?

Corrections:

1.     Lines 59 -62. This sentence needs rewording: ".....previously investigated, except of apart from our study [18] where we in which we observed a higher..."

2.     Lines 96 & 97: Only the stub of the aorta was "retrogradely perfused", and NOT the coronary circulation. Please state this clearly.

3.     Lines 281-282. Wording needs to be clarified. Meant to state that alternans was absent in the AIP treated cells.

4.     The Y axis label in Figure 3 B d needs rewording: They are not "alternating cells", but "cells with alternans".

5.     Can you confirm the same stimulation frequency was used for figure 4 A a and 4 B a? The "1 s" time bar looks to be shorter in B a.
